



Earth System
Science
Data

# The TROPOSIF global sun-induced fluorescence dataset from the Sentinel-5P TROPOMI mission

Luis Guanter[1], Cédric Bacour[2], Andreas Schneider[3,a], Ilse Aben[3], Tim A. van Kempen[3],
Fabienne Maignan[4], Christian Retscher[5], Philipp Köhler[6], Christian Frankenberg[6], Joanna Joiner[7], and
Yongguang Zhang[8]

[1]Research Institute of Water and Environmental Engineering (IIAMA),
Universitat Politècnica de València, Valencia, Spain
[2]NOVELTIS, Labège, France
[3]SRON Netherlands Institute for Space Research, Utrecht, the Netherlands
[4]Laboratoire des Sciences du Climat et de l'Environnement (LSCE), Institut Pierre Simon Laplace (IPSL),
Gif-sur-Yvette, France CE1
[5]European Space Agency (ESA/ESRIN), Frascati, Italy
[6]Jet Propulsion Laboratory, California Institute of Technology, Pasadena, California, USA
[7]Goddard Space Flight Center (GSFC), National Aeronautics and Space Administration (NASA),
Greenbelt, MD, USA
[8]International Institute for Earth System Science, Nanjing University, Nanjing, Jiangsu, China
[a]now at: Finnish Meteorological Institute, Sodankylä, Finland TS1

**Correspondence:** Luis Guanter (lguanter@fis.upv.es)

**Abstract.** CE2 CE3 The first satellite-based global retrievals of terrestrial sun-induced chlorophyll fluorescence (SIF) were achieved in 2011. Since then, a number of global SIF datasets with different spectral, spatial, and temporal sampling characteristics have become available to the scientific community. These datasets have been useful to monitor the dynamics and productivity of a range of vegetated areas worldwide, but the coarse spatiotemporal sampling and low signal-to-noise ratio of the data hamper their application over small or fragmented ecosystems. The recent advent of the Copernicus Sentinel-5P TROPOMI mission and the high quality of its data products promise to alleviate this situation, as TROPOMI provides daily global measurements at a much denser spatial and temporal sampling than earlier satellite instruments. In this work, we present a global SIF dataset produced from TROPOMI measurements within the TROPOSIF project funded by the European Space Agency. The current version of the TROPOSIF dataset covers the time period between May 2018 and April 2021. Baseline SIF retrievals are derived from the 743–758 nm window. A secondary SIF dataset derived from an extended fitting window (735–758 nm window) is included. This provides an enhanced signal-to-noise ratio at the expense of a higher sensitivity to atmospheric effects. Spectral reflectance spectra at seven 3 nm windows devoid of atmospheric absorption within the 665–785 nm range are also included in the TROPOSIF dataset as an important ancillary variable to be used in combination with SIF. The methodology to derive SIF and ancillary data as well as results from an initial data quality assessment are presented in this work. The TROPOSIF dataset is available through the following digital object identifier (DOI): https://doi.org/10.5270/esa-s5p_innovation-sif-20180501_20210320-v2.1-202104 (Guanter et al., 2021).

# 1 Introduction

The sun-induced fluorescence (SIF) signal emitted by the chlorophyll *a* of terrestrial vegetation has been shown to be a closer indicator of vegetation functioning than other variables traditionally derived from optical remote sensing data (Mohammed et al., 2019). Global retrievals of SIF from space were first achieved in late 2011 from the GOSAT mission (Frankenberg et al., 2011b; Joiner et al., 2011; Guanter et al., 2012). Since then, a number of global SIF datasets have been produced from spaceborne spectrometers originally intended for atmospheric research, such as GOME-2 (e.g., Joiner et al., 2013; Köhler et al., 2015a; van Schaik, 2020), SCIAMACHY (Köhler et al., 2015a; Khosravi et al., 2015; Joiner et al., 2016), OCO-2 (e.g., Frankenberg et al., 2014; Sun et al., 2018), and TanSat (e.g., Du et al., 2018; Yao et al., 2021). The SIF data from those missions have been typically used to investigate the spatial and temporal patterns of the gross primary production (GPP) of different ecosystems (e.g., Frankenberg et al., 2011b; Guanter et al., 2014; Yoshida et al., 2015; Sun et al., 2017; Luus et al., 2017; Walther et al., 2017; Jeong et al., 2017; Smith et al., 2018; Zuromski et al., 2018; Wu et al., 2018) and also ecosystem transpiration and water limitation (Pagán et al., 2019; Maes et al., 2020; Shan et al., 2021).

The main limitations of those SIF datasets for scientific work are the high precision errors and the low spatial resolution and/or sparse spatial sampling. ESA's FLuorescence EXplorer mission, scheduled for launch after 2025, will enable a breakthrough in the spatial resolution of space-based SIF datasets (Drusch et al., 2017). Before that, the advent of the TROPOspheric Monitoring Instrument (TROPOMI) aboard the Copernicus Sentinel-5P mission in October 2017 is already helping to substantially reduce those limitations in spatiotemporal sampling and signal-to-noise ratio. TROPOMI combines a global continuous spatial sampling with a $3.5 \times 7.5\,\mathrm{km}^2$ pixel size at nadir in the near infrared ($3.5 \times 5.5\,\mathrm{km}^2$ since August 2019) with a daily revisit time, which leads to a large increase in the number of clear-sky measurements per day in comparison to earlier missions. In addition, it measures with a high signal-to-noise ratio, a spectral resolution of $0.37\,\mathrm{nm}$, and a wide spectral coverage in the near-infrared window. These all enable high-performance SIF retrievals (see Fig. 1). The high potential of TROPOMI for SIF monitoring was first anticipated by the sensitivity analysis performed by Guanter et al. (2015) and later demonstrated by the first SIF retrievals from real TROPOMI data recently published by Köhler et al. (2018) and Köhler et al. (2020) (hereinafter referred to as the "Caltech product"). The first publications exploiting TROPOMI SIF data for scientific applications confirm those high expectations on the use of TROPOMI for vegetation monitoring (Turner et al., 2020; Doughty et al., 2019; Zhang et al., 2019; Yin et al., 2020; He et al., 2020).

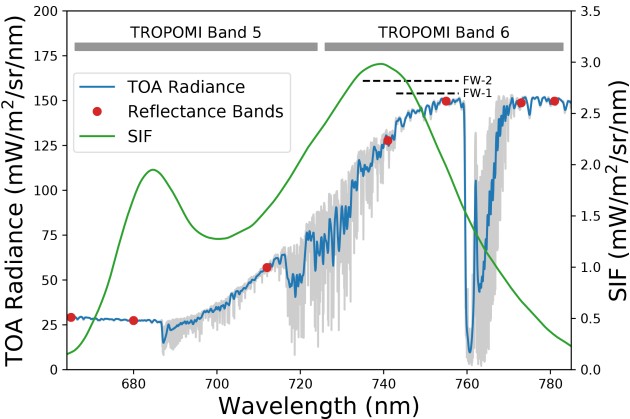

**Figure 1.** Top-of-atmosphere (TOA) radiance spectrum and top-of-canopy SIF spectrum from a green vegetation surface in the 665–785 nm spectral range covered by TROPOMI's bands 5 and 6. The TOA radiance spectrum at high spectral resolution is plotted in gray, and the result of the convolution with TROPOMI's spectral resolution is shown in blue. The horizontal dashed lines show the span of the baseline SIF retrieval fitting window (FW-1, 743–758 nm) and the secondary fitting window (FW-2, 735–758 nm). The spectral locations of the 3 nm macro-channels at which spectral reflectance data are provided in the TROPOSIF product are marked in red. A typical SIF spectrum is depicted in green.

Regarding the exploitation of SIF data for scientific applications, the so-called canopy structure effect has to be taken into account either to disentangle the physiological information in SIF from other confounding variables, to normalize the impact of illumination–observation geometries, or to quantify GPP from SIF. The structure of the canopy defines wavelength-dependent multiple scattering and absorption processes, which determine the amount of SIF photons escaping the vegetation cover and hence the SIF flux measured at the satellite level (Joiner et al., 2020). A number of studies are proposing the use of spectral surface reflectance in the visible (especially, wavelengths in the red) to near-infrared parts of the spectrum to help model those radiative transfer processes (Yang and van der Tol, 2018; Liu et al., 2019; Zhang et al., 2020; Badgley et al., 2017; Dechant et al., 2020; Yang et al., 2020). In addition, spectral reflectance from the entire 675–775 nm range can be used to derive indicators of the fraction of absorbed photosynthetically active radiation (FAPAR), leaf-area index (LAI), and other biophysical parameters and spectral vegetation indices. These can complement SIF for the characterization of the vegetation condition and functioning. For example, it has been recently shown that the near-infrared reflectance of vegetation (NIRv) index, or its variation resulting from the multiplication by incoming sunlight (NIRvP), is a good proxy for SIF (Badgley et al., 2017; Dechant et al., 2020, 2021). NIRv and NIRvP could then provide useful information for the calculation of the SIF yield, which is a top-of-canopy variable providing information on both fluorescence yield (leaf-level variable)

and any remaining canopy structure effects not accounted for by NIRv and NIRvP. The fact that 675–775 nm spectral reflectance can be derived from the same TROPOMI spectra as SIF (see Fig. 1) and that this reflectance data are provided at a sufficiently high spatial resolution (e.g., similar to some MODIS global vegetation products) is another advantage of using TROPOMI data for vegetation monitoring (Guanter et al., 2015).

In this paper, we describe the SIF and spectral reflectance data product derived from TROPOMI in the frame of the TROPOSIF project funded by the Sentinel-5p+ Innovation activity of the European Space Agency (ESA)[1]. Our work is aimed at developing a TROPOMI-based SIF processor which can be implemented at ESA's data processing facilities for the operational generation and distribution of the data product to users. In addition to SIF, reflectance spectra from each input radiance spectrum are also included in the product for combination with the SIF retrievals.

## 2 Methodology

### 2.1 SIF retrieval approach

The different SIF retrieval schemes implemented for spaceborne measurements in the last years have relied on the Fraunhofer line in-filling principle proposed by Plascyk and Gabriel (1975). This principle establishes that the fractional depth of solar Fraunhofer lines, which are included in the reflected solar light spectrum, decreases when it is combined with an additive signal such as SIF. The feasibility of this concept for satellite-based SIF retrieval was first shown by simulations (Sioris et al., 2003; Frankenberg et al., 2011a) and then applied to all satellite missions for which global SIF data products have been derived.

We can group SIF retrieval strategies into physically based and data-driven methods. The first group of methods are based on a physical formulation of the radiative transfer and the instrument's response for the modeling of top-of-atmosphere (TOA) radiance measurements. These types of methods have been applied to instruments with a high spectral resolution, such as GOSAT and OCO-2 (Frankenberg et al., 2011b; Köhler et al., 2015b). In turn, data-driven methods model the input TOA radiance spectra as a linear combination of spectral functions derived from statistical analysis of SIF-free training sets. The performance of these methods is conditioned by the arbitrary selection of the training set and several model parameters, but they are fast and very effective in accounting for both atmospheric and instrumental effects. Data-driven methods have been used with both high- and medium-spectral-resolution instruments (Guanter et al., 2012; Joiner et al., 2013; Köhler et al., 2015a; Joiner et al., 2016; Sanders et al., 2016; Guanter et al., 2015; Köhler et al., 2018; van Schaik et al., 2020).

In this work, we have adapted the data-driven retrieval scheme described in Guanter et al. (2015) for the processing of real TROPOMI data. The forward model takes the form

$$F(\boldsymbol{a}, \boldsymbol{\alpha}, F_s) = \boldsymbol{v}_1 \sum_{i=0}^{n_p} a_i \boldsymbol{\lambda}^i + \sum_{j=2}^{n_v} \alpha_j \boldsymbol{v}_j + F_s \boldsymbol{h}_F, \tag{1}$$

where $\boldsymbol{\lambda}$ [TS3] is the array of measurement wavelengths used for the representation of spectrally smooth terms such as surface reflectance and atmospheric scattering; $\boldsymbol{v}$ is the basis of principal components describing the variability in solar irradiance and atmospheric transmittance; $\boldsymbol{a}$ represents the coefficients of a polynomial in wavelength; $\boldsymbol{\alpha}$ represents the weights of the singular vectors; $F_s$ is SIF at the reference wavelength of 740 nm; $\boldsymbol{h}_F$ is a fixed spectral function normalized at 740 nm, which accounts for the spectral shape of SIF; $n_p$ is the order of the polynomial used to represent spectrally smooth terms; and $n_v$ is the number of principal components representing high-spectral-frequency variations. The first two terms on the right hand side of Eq. (1) are a simplification of the product of $\sum_{i=0}^{n_p} a_i \boldsymbol{\lambda}^i$ and $\sum_{j=1}^{n_v} \alpha_j \boldsymbol{v}_j$, which are terms representing contributions of low and high spectral frequencies to the reflected solar radiation, respectively (Guanter et al., 2013, 2015). The third term on the right hand side of Eq. (1) is the SIF contribution to the TOA radiance measurement. The absorption of SIF by the atmosphere between the ground and the TOA can be neglected for fitting windows devoid of strong atmospheric absorption lines such as the ones selected in this work (see Sect. 2.2), which was justified in Guanter et al. (2015). The effect of atmospheric absorption on SIF retrievals at far-red wavelengths had been previously evaluated by means of simulation in Guanter et al. (2012); Frankenberg et al. (2012). The effect should be in the range $\sim 3\%$–$6\%$ for a typical aerosol optical thickness of 0.2 and observation angles between 0 and 45°.

The state vector elements to be estimated in the retrieval process are $a_i$, $\alpha_j$, and $F_s$, whereas $\boldsymbol{\lambda}$, $\boldsymbol{v}$, and $\boldsymbol{h}_F$ are model parameters. Regarding these, $\boldsymbol{\lambda}$ is known for each input TOA spectrum. The singular vectors $\boldsymbol{v}$ are calculated through singular vector decomposition of a training set consisting of TOA radiance spectra extracted from measurements over non-vegetated areas (mostly, the Sahara desert and Arctic and Antarctic surfaces). The training is done on a per-column basis in order to account for slight variations in the spectral and radiometric response of different detectors in the across-track direction. Finally, the spectral function $\boldsymbol{h}_F$ is extracted from spectral libraries (Guanter et al., 2013). The forward model is inverted for each TOA radiance spectrum using ordinary least squares.

### 2.2 Retrieval fitting windows

We have chosen to run the TROPOSIF retrieval for two fitting windows, 743–758 and 735–758 nm. The first one is purely based on solar Fraunhofer lines, whereas the second is

[1]https://eo4society.esa.int/projects/sentinel-5pinnovation/ [TS2]

also slightly affected by water vapor lines in the 735–743 nm range (see Fig. 1). These two windows are selected as good compromises between retrieval noise and sensitivity to cloud contamination, as it was demonstrated in the sensitivity analysis by Guanter et al. (2015): the 735–758 nm window leads to smaller precision errors than the narrower 743–758 nm window because of the larger number of spectral points, whereas the 743–758 nm window is more robust against atmospheric effects due to the absence of atmospheric lines. The 743–758 nm window is therefore a better choice for applications not requiring strict clear-sky observations, as the impact of sub-pixel clouds is lower than for the other window and the greater number of observations available compensate for the higher retrieval random errors; for applications only relying on clear-sky data, the 735–758 nm window would be a better option, as the lower precision errors would compensate for the lower number of available measurements and the higher sensitivity to sub-pixel clouds would not be relevant.

The number of singular vectors $n_v$ is set to four for the 743–758 nm fitting window and seven for 735–758 nm. This choice is based on the results from the sensitivity analysis by Guanter et al. (2015) and recent tests with real TROPOMI data. Results from those tests can be seen in Fig. 2, which shows an abrupt drop in the weight of the $v$ after four for the 743–758 nm window and seven for the 735–758 nm window. On the other hand, $n_p$ is set to three in both cases. We have not found major differences in the retrieval accuracy and precision when using a greater $n_v$ or $n_p$.

An example of the three first $v_i$ and their weights for the 743–758 nm fitting window derived from the singular vector decomposition of a particular training set are displayed in Fig. 2. It can be observed that $v_1$ carries most of the weight (i.e., reproduces most of the variance of the training set), whereas the others may account for low-spectral-frequency variations in albedo (e.g., $v_2$) or measurement artifacts like spectral shifts (e.g., $v_3$). This justifies the choice of only multiplying $v_1$ by the polynomial in wavelength accounting for low-frequency spectral variations in Eq. (1).

## 2.3 Retrieval random error

As it was described in Guanter et al. (2015), the retrieval error covariance matrix $\mathbf{S}_e$ is given by

$$\mathbf{S}_e = \left(\mathbf{J}^T \mathbf{S}_0^{-1} \mathbf{J}\right)^{-1}, \tag{2}$$

where $\mathbf{S}_0$ is the measurement error covariance matrix and $\mathbf{J}$ the Jacobian matrix, which in the particular case of the retrieval forward model given in Eq. (1) consists of the terms $\mathbf{J}(a_i) = v_1 \lambda^i$, $\mathbf{J}(\alpha_j) = v_j$ and $\mathbf{J}(F_s) = h_F$. The last element in the diagonal of the $\mathbf{S}_e$ matrix contains the squared random error of the retrieved SIF value.

As a result, the retrieval random error for $F_s$ only depends on measurement noise (i.e., the noise in the input radiance spectra), which in turn is driven by incoming radiance

through photon noise. For the robust and computationally efficient calculation of retrieval errors, empirical radiance–$\sigma(\text{SIF})$ curves have been produced for each fitting window. The $1\sigma$ retrieval errors were first calculated by means of Eq. (2) for several orbits and grouped into TOA radiance bins of $5\,\text{mW}\,\text{m}^{-2}\,\text{sr}^{-1}\,\text{nm}^{-1}$. A third-order polynomial was then fitted to the binned radiance–$\sigma(\text{SIF})$ data. The result of this process is shown in Fig. 3. The resulting polynomial coefficients are used to predict $\sigma(\text{SIF})$ as a function of average radiance on a per-pixel basis. With this strategy we avoid the inversion of the measurement error covariance matrix in Eq. (2), which had proved to be unstable and computationally expensive. It must be mentioned that the measurement noise data provided in TROPOMI's Level 1B (L1B) files suffer from float number truncation errors from the storage as byte data. This means that the $\sigma(\text{SIF})$ that we calculate from the measurement noise data can affected by the same effect. This manifests as a underestimation of the $\sigma(\text{SIF})$ values of a varying magnitude (from 0 % to $\sim 15\,\%$ TS4) along the radiance–$\sigma(\text{SIF})$ curve.

## 2.4 Day-length scaling

The variability in SIF retrievals due to instantaneous illumination conditions must be accounted for when SIF retrievals are to be compared with other SIF datasets acquired with a different satellite overpass time or with daily GPP estimates (Zhang et al., 2018).

We have followed the approach proposed by Frankenberg et al. (2011b) to generate daily-equivalent SIF (SIF$_d$) estimates in TROPOSIF. This is based on the calculation of a day-length scaling factor DL as

$$\text{DL}(t_0) = \frac{\int_{t_{\text{sr}}}^{t_{\text{ss}}} \mu_s(t)\,\text{d}t}{\mu_s(t_0)}, \tag{3}$$

where $t_0$ is the time of the measurement, $t_{\text{sr}}$ is the time of sunrise, $t_{\text{ss}}$ is the time of sunset, and $\mu_s(t)$ is the cosine of the sun zenith angle. This approach assumes that SIF scales linearly with incoming solar radiation and that the entire day is cloud-free.

## 2.5 Quality value for SIF retrievals

An empirical quality value (`qa_value`) indicating the reliability of each SIF retrieval is included in the TROPOSIF product. This value is a score between 0 and 1 (from lowest to highest quality) calculated as the combination of a series of empirical penalty factors applied in those conditions which are expected to potentially degrade the retrieval quality.

Starting from a maximum `qa_value` of 1.0, penalty factors are applied on the view zenith angle (VZA), the sun zenith angle (SZA), and the reduced $\chi^2$ of the fit ($\chi_r^2$) as follows:

1. VZA $> 60° \rightarrow$ `qa_value` = `qa_value` $- 0.5$;

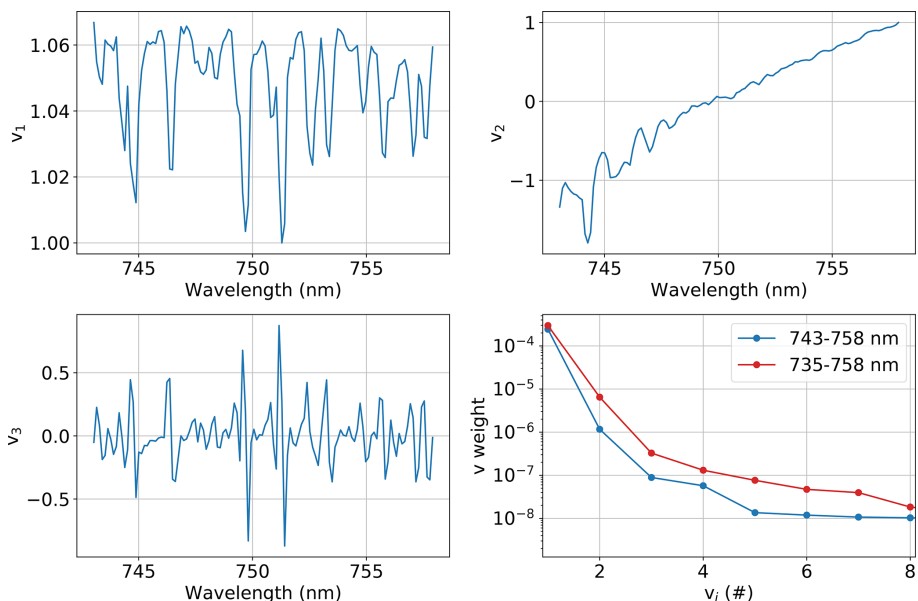

**Figure 2.** First three singular vectors ($\mathbf{v}_i$) and weights of the first eight $\mathbf{v}_i$ for the 743–758 fitting window from the singular vector decomposition of a training set consisting of TROPOMI spectra over non-vegetated areas. The weights of the first eight $\mathbf{v}_i$ are also shown for the 735–758 nm fitting window.

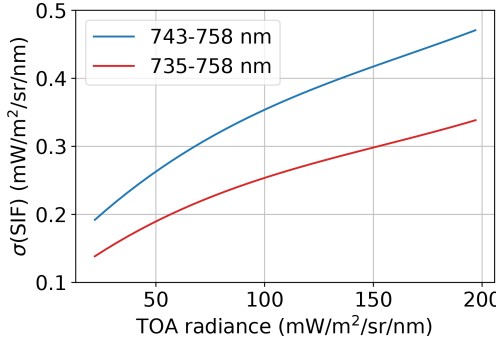

**Figure 3.** Empirical radiance–$\sigma$(SIF) curves used for the calculation of $1\sigma$ SIF retrieval errors in the operational processing.

2. SZA $> 70° \rightarrow$ qa_value = qa_value − 0.5;

3. average TOA radiance $\notin [20,\ 200]\,\mathrm{mW\,m^{-2}\,sr^{-1}\,nm^{-1}}$ $\rightarrow$ qa_value = qa_value − 0.5;

4. $\chi_r^2 \notin [0.6,\ 2] \rightarrow$ qa_value = qa_value − 1.0;

5. SIF $\notin [-10,\ 10]\,\mathrm{mW\,m^{-2}\,sr^{-1}\,nm^{-1}} \rightarrow$ qa_value = qa_value − 1.0.

If the final qa_value becomes $< 0$, it is reset to 0.

Cloud contamination is not part of the computation of the qa_value. Cloudy observations are included in the generation of the singular vectors used by the forward model, so the latter is generally expected to properly represent cloudy spectra. In those cases in which clouds do degrade the fit quality, this will be captured in the $\chi_r^2$ value. Further filtering

by cloud fraction is possible for those applications in which only cloud-free acquisitions are used.

Only SIF retrievals with qa_value = 1.0 (equivalently, qa_value $> 0.5$) are recommended for use. In practice, this means that the data point should be discarded if any of the previous conditions applies.

## 2.6 Spectral reflectance

Spectral reflectance spectra at seven spectral points in the 665–785 nm range covered by TROPOMI's band 5 and 6 (namely, 665, 680, 712, 741, 755, 773, and 781 nm) are included in the TROPOSIF product as an ancillary dataset. Spectral reflectance spectra are calculated as

$$\rho_{\mathrm{TOA},i} = \frac{\pi \langle L_{\mathrm{TOA}}\rangle_i}{\mu_s \langle I_{\mathrm{sc}}\rangle_i}, \tag{4}$$

where $L_{\mathrm{TOA}}$ refers to the TOA radiance measurement (L1B radiance spectra); $I_{\mathrm{sc}}$ is an extraterrestrial solar irradiance spectrum corrected by the Sun-to-Earth distance at the day of the overpass; and $\langle\rangle_i$ refers to the spectral convolution operator, which in this case corresponds to a boxcar spectral response function with a 3 nm width, applied at the $i$th spectral point. The Kurucz 2005 [TS5] high-resolution irradiance spectrum[2] (from 300 to 1000 nm, with $\sim 0.001$ nm sampling at 740 nm) resampled at the seven spectral points with the same 3 nm boxcar function is used for $\langle I_{\mathrm{sc}}\rangle$. This approach is computationally much simpler than the alternative of using

---

[2]http://kurucz.harvard.edu/sun/irradiance2005/irradthu.dat [TS6]

TROPOMI's solar irradiance product and still yields a sufficient accuracy because of the reduced spectral resolution of the $\rho_{\text{TOA},i}$ calculations.

The seven spectral points are selected so that they provide both a proper sampling of the red, red-edge, and near-infrared parts of the spectrum and a low sensitivity to atmospheric effects (see Fig. 1). This is important because no atmospheric correction is performed on the spectral reflectance spectra.

## 2.7   Processing flow

TROPOSIF's processing chain starts by reading in TOA radiance data from TROPOMI band 5 and 6 L1B orbit files as well as cloud fraction data from the TROPOMI L2 Cloud product. Data acquired over water bodies (identified using the MODIS MCD12C1 2018 land cover product) with a high cloud fraction ($> 0.8$) or with a lower quality according to the `quality_level` flag attached to the L1B data (threshold of 80) are screened out from the processing. Spectral band no. 179 in TROPOMI's channel 6 is also removed from the processing, since spectral spikes were found for that band in pixels in the vicinity of clouds.

The first step in the processing is the generation of the singular vectors. For that, spectra over barren areas worldwide as identified in the MODIS MCD12C1 2018 dataset are used. Next, TOA radiance data from either a continuous fitting window (for SIF retrieval) or at certain "macro-channels" at atmospheric windows within the far-red region (for reflectance retrieval) are read in. SZA and latitude and longitude fields are also extracted for the computation of the DL factor. SIF and reflectance retrievals are run separately for each column in order to account for column-dependent changes in TROPOMI's spectral and radiometric responses. The output SIF, spectral reflectance, and DL are written to a NetCDF file together with other variables of interest for later data visualization and processing. This NetCDF file has the same structure as the official TROPOMI L2 products (see Appendix A).

## 2.8   L2B processing

In order to facilitate data download and processing, a final processing step extracts all valid retrievals (`qa_value >` 0.5) from the L2 orbit files in a given day and combines them into single daily files, referred to as L2B files. The application of the `qa_value` filter substantially reduces the volume of data in the L2B file with respect to the combined L2 dataset. In the case of spectral reflectance data, only spectra acquired under a cloud fraction smaller than 0.2 are included. For the sake of reducing the data volume, some information fields in the L2 files are not included in L2B files, such as the $\chi_{\text{r}}^2$ values, the `qa_value`, the day-length factor, and the illumination and observation azimuth angles (which are converted into a single relative azimuth angle).

## 3   Results

### 3.1   Consistency of SIF retrievals

Maps from spatial subsets of different variables derived from the processing of one TROPOMI orbit dataset are displayed in Fig. 4 TS7. The different panels show the spatial distribution of TOA radiance at 743 nm, the cloud fraction in TROPOMI's L2 Cloud product, `qa_value`, the TOA reflectance at 665 and 781 nm, the normalized difference vegetation index (NDVI) derived from those two channels, and the retrieved SIF, $1\sigma$ error, and the $\chi_{\text{r}}^2$ of the fit for both 743–758 and 735–758 nm fitting windows. $\chi_{\text{r}}^2$ values larger (smaller) than 1 indicate an underestimation (overestimation) of the model–data error variance.

The following specific points can be highlighted from Fig. 4.

- The areas with the highest abundance of green vegetation (e.g., the northern Iberian Peninsula, South West France, Brittany, and Normandy) are clearly visible in the SIF and NDVI maps. In the case of SIF, the range of values is within the expectation for both fitting windows. Slightly higher values are found for the 743–758 nm fitting window in clear-sky regions.

- Single-retrieval uncertainties are also within the expected range (0.3–0.6 mW m$^{-2}$ sr$^{-1}$ nm$^{-1}$). They are substantially lower for the 735–758 nm window.

- For both fitting windows, the $\chi_{\text{r}}^2$ is typically around 0.8–0.9. This deviation from an ideal case of $\chi_{\text{r}}^2 = 1$ would indicate that the model is slightly over-fitting the data ($\chi_{\text{r}}^2 > 1$ implies that the model does not model the full variance in the spectrum). However, the 0 %–15 % truncation errors in the measurement noise input data discussed in Sect. 2.3 could also explain that $\chi_{\text{r}}^2$ is in general smaller than 1. On the other hand, the sensitivity to clouds of the 735–758 nm retrieval can also be noticed in the $\chi_{\text{r}}^2$ maps. Relatively high $\chi_{\text{r}}^2$ values ($> 1.5$) are found for retrievals over cloudy areas for 735–758 nm retrievals, which is explained by the existence of water vapor absorption lines in the 735–743 nm range. No relevant differences in $\chi_{\text{r}}^2$ between clear and cloudy areas are found for the 743–758 nm window. More details on the fit performance can be found in Fig. 5.

- The quality value map indicates that the retrievals are in general performed under the best retrieval conditions (`qa_value = 1.0`). Lower-quality retrievals are found for the edges of the subset where VZA $> 60°$, for some patches in the Sahara desert where the mean TOA radiance is greater than the 200 mW m$^{-2}$ sr$^{-1}$ nm$^{-1}$ threshold, and for the cloudy areas on the top right of the sampled area for which $\chi_{\text{r}}^2 > 2$.

- Even though the processing is performed on a per-column basis, detector artifacts in the form of striping

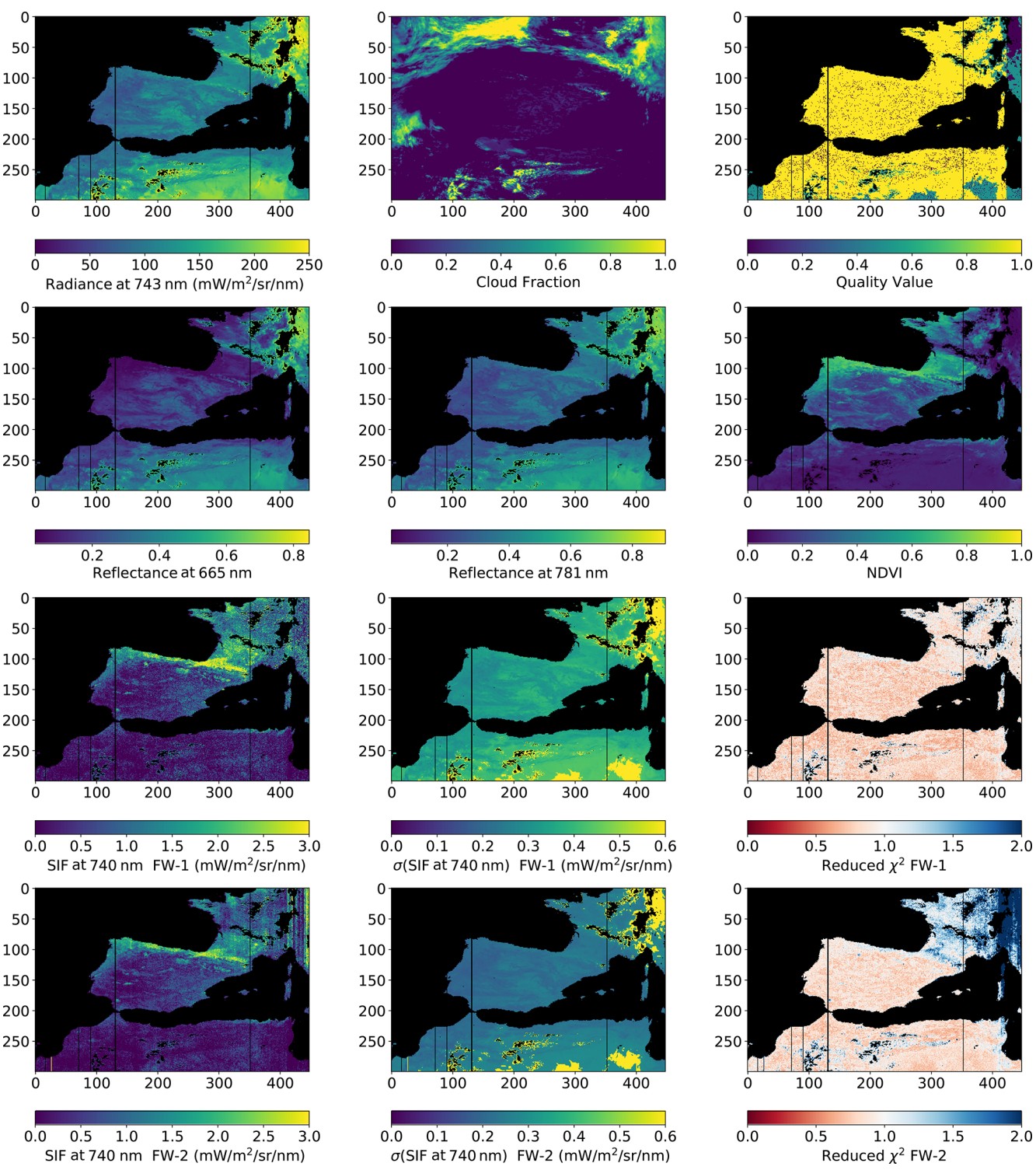

**Figure 4.** Subsets of the input data (TOA radiance at 743 nm and cloud fraction) and the results obtained from the processing of TROPOMI's 09025 orbit (11 July 2019) with the TROPOSIF SIF processor. FW-1 refers to the 743–758 nm fitting window and FW-2 to 735–758 nm.

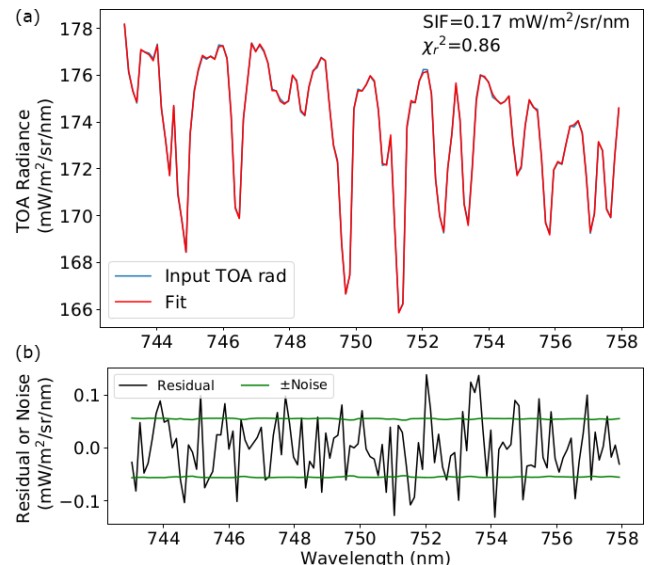

**Figure 5.** Example of spectral fit for the 743–758 nm fitting window. The fit residual is compared to the noise level included in the input L1B dataset for the same spectrum. The resulting SIF values at 740 nm and $\chi_r^2$ of the fit are shown in panel **(b)**. TS8

are present in the SIF maps, which can be explained by an across-track dependence of TROPOMI's spectral and radiometric calibration. In particular, the processing has not run over some across-track columns which did not pass the `quality_level > 80` filter, and higher uncertainty values are found at the swath edges.

## 3.2  Spatiotemporal composites of the all-sky SIF product

A global composite of SIF data from the 743–758 nm fitting window in a 0.2° latitude–longitude grid for the 8–15 July 2019 period and a cloud fraction lower than 0.5 is presented in Fig. 6. A total of 102 orbit files have been used for the composite. Prior to cloud filtering, retrievals are filtered using the recommended `qa_value > 0.5`, which corresponds to VZA < 60°, average TOA radiance between 20 and 200 mW m$^{-2}$ sr$^{-1}$ nm$^{-1}$, and $\chi_r^2$ between 0.6 and 2.5 for consistency with the filtering scheme proposed by Köhler et al. (2018). The number of retrievals used in each 0.2° grid box and the resulting standard error of the mean ($\sigma$), calculated as described in Guanter et al. (2015), are shown in the smaller panels of Fig. 6.

The resulting 8–15 July 2019 SIF map shows the expected SIF global maximum at the US Corn Belt (Guanter et al., 2014). High SIF values are also found in Central America, Central Asia, and China. The data gap in the southern part of Greenland is due to the incoming radiance being higher than the 200 mW m$^{-2}$ sr$^{-1}$ nm$^{-1}$ threshold used for quality filtering, and the one in southern Argentina is due to SZA > 70°. The effect of cloud filtering on the stan-

dard errors of SIF composites is shown in the $\sigma$(SIF) map. It is especially noticeable in regions with persistent cloud cover such as the tropics and the northern part of Europe and America. Overall, typical standard errors are in the range 0.1–0.2 mW m$^{-2}$ sr$^{-1}$ nm$^{-1}$ for this weekly composite.

The impact of cloud fraction thresholds on SIF composites (systematic and random errors) has been further investigated. Figure 7 displays the differences in SIF and $\sigma$(SIF) composites (8–15 July 2019, 743–758 nm fitting window) using either a strict or a relaxed cloud fraction threshold (0.2 and 0.8, respectively). As expected, the absolute SIF values increase with the strictness of the cloud filter, as clouds have a shielding effect for SIF photons traveling from the surface to the sensor. At the same time, the standard errors of SIF composites decrease with the cloud fraction threshold following the increase in the number of retrievals per grid box. For the selected 8–15 July 2019 time period, the greatest impact of clouds in SIF and $\sigma$(SIF) is found in Western Europe. In this region, the change from 0.8 to 0.2 cloud fraction thresholds implies an increase in the SIF average of about 0.4–0.6 mW m$^{-2}$ sr$^{-1}$ nm$^{-1}$ at the grid box level, which can be understood as a systematic error in the SIF composite due to clouds. For the same region, the change of the standard error due to the lower number of retrievals in the stricter cloud filtering is about 0.25 mW m$^{-2}$ sr$^{-1}$ nm$^{-1}$. This illustrates that the selection of the optimal cloud fraction threshold depends on the particular use of the data. A strict cloud filtering (e.g., cloud fraction lower than 0.2) should be applied if an accurate SIF mean value is needed, whereas a more relaxed cloud filter (e.g., cloud fraction lower than 0.8) could be applied if smooth spatial and/or temporal signals are needed. The second case is often preferred for SIF, but it must be taken into account that unfiltered clouds can also introduce noise-like changes in SIF time series. Including cloud fraction data in the analysis would be needed for a proper interpretation of the SIF signals if a relaxed cloud filtering was applied.

The seasonal variation of TROPOSIF SIF data is illustrated in Fig. 8, which shows daily and 0.1° composites over a set of instrumented SIF sites (Parazoo et al., 2019) and the entire time period covered by the TROPOSIF product. Two configurations of fitting window and cloud fraction thresholds are used: 743–758 nm with cloud fraction < 0.8 ("all sky") and 735–758 nm with a cloud fraction < 0.2 ("clear sky"). The higher number of daily estimates available for the all-sky case is a result of less strict cloud filtering. Figure 8 illustrates the lower retrieval uncertainty associated with the clear-sky case (lower error bars and less scattering) compared to the all-sky SIF. SIF data derived from the 743–758 nm fitting window are usually greater than those derived from the broader 735–758 nm window, as indicated by the slope of the linear fit in the scatterplots on the right hand side of Fig. 8. On the other hand, negative SIF values are found for the two fitting windows. These are caused by noise in the data being propagated to random errors in the retrieved SIF.

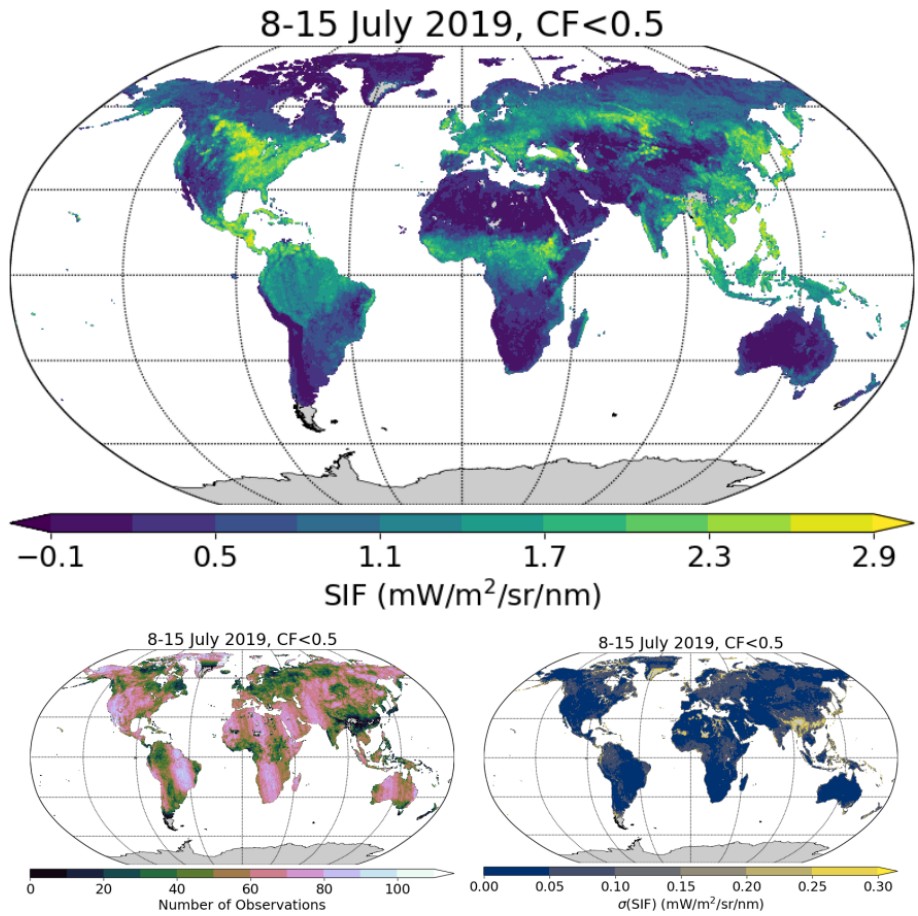

**Figure 6.** Global composites of SIF at 740 nm for the time period 8–15 July 2019 using retrievals from the 743–758 nm fitting window and a cloud fraction (CF) threshold of 0.5. Data are gridded in 0.2° grid boxes. The number of observations and the resulting standard error of the mean per grid box are shown in the lower panels.

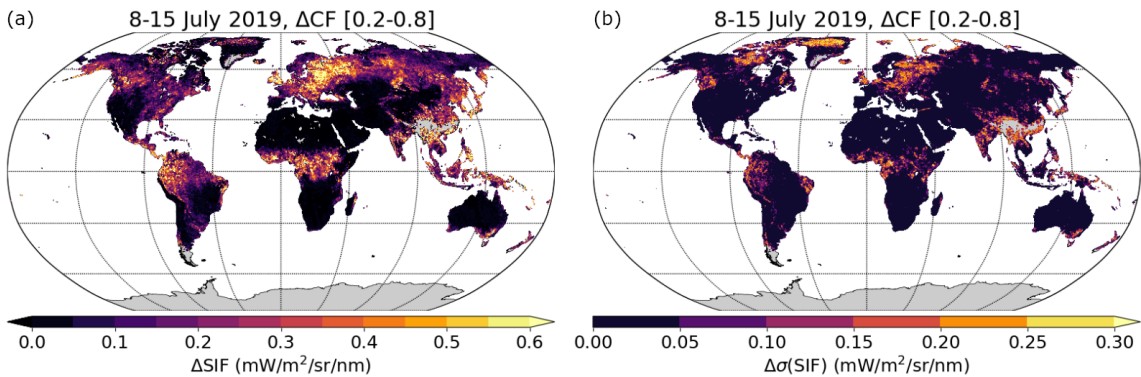

**Figure 7.** Differences of average SIF **(a)** and $\sigma$(SIF) **(b)** for 0.2 and 0.8 cloud fraction thresholds. Data are from the time period 8–15 July 2019 and the 743–758 nm fitting window.

Removing negative SIF retrievals from the processing should be avoided, as it would lead in positively biased averages.

It must be stated that the slopes of the scatterplots in Fig. 8 cover the minimum and maximum slope values found in the analysis of 13 different sites (roughly, 1.0 to 1.5). Further re-

search is needed to understand whether the variations in the slopes are due to retrieval biases over some vegetation types or to leaf/canopy radiative transfer effects making the shape of the SIF emission dependent on the leaf and canopy type. A dependence of SIF retrievals on the fitting window and atmo-

https://doi.org/10.5194/essd-13-1-2021 Earth Syst. Sci. Data, 13, 1–18, 2021

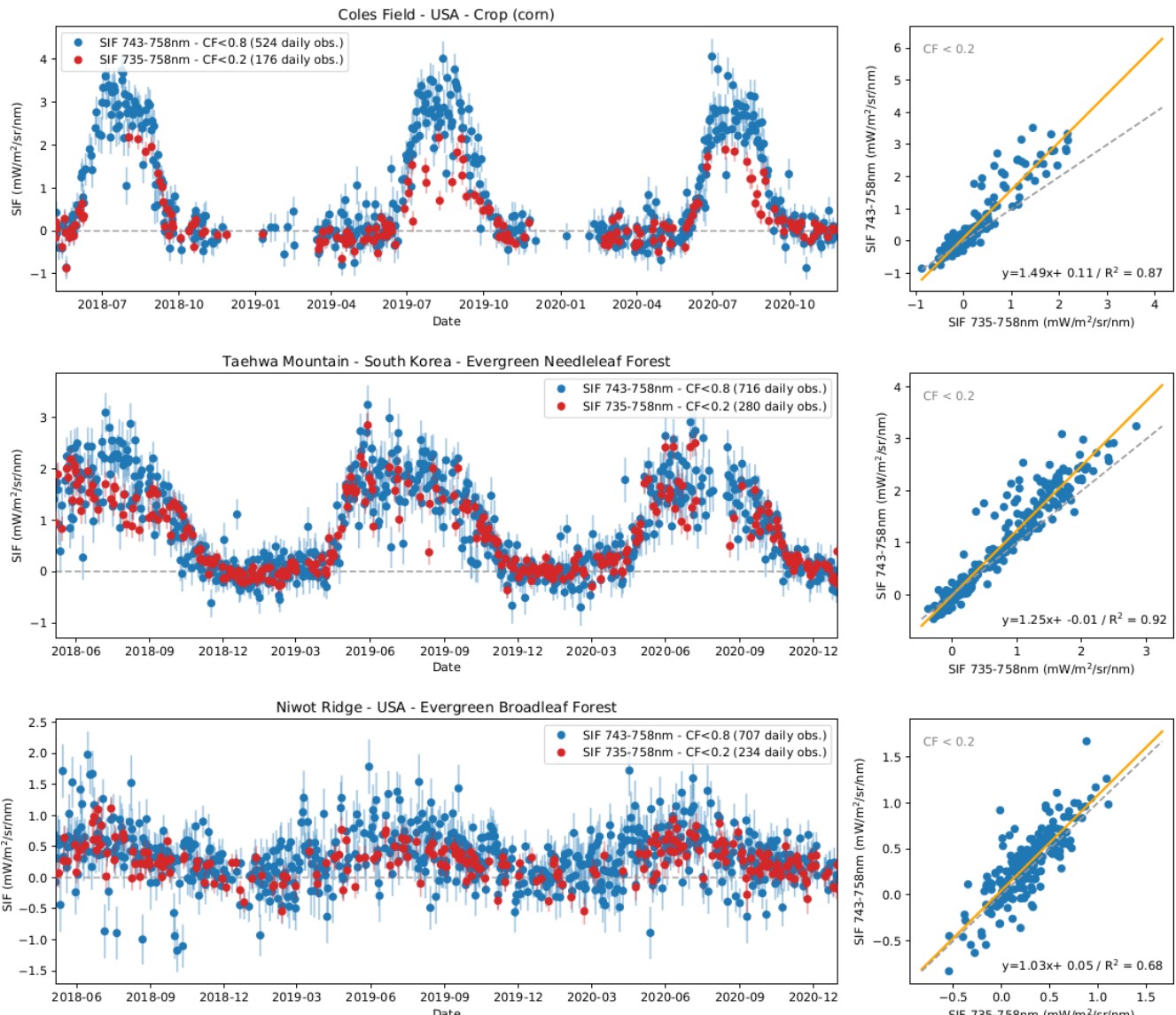

**Figure 8.** Comparison of SIF retrievals from the 743–758 and 735–758 nm fitting windows and 0.2 and 0.8 cloud fraction (CF) thresholds at three sites (0.1° radius) corresponding to different biomes (Parazoo et al., 2019). Left: time series of the daily SIF retrievals for two combinations of fitting window and cloud screening (743–758 nm data with CF < 0.8 and 735–758 nm with CF < 0.2) from May 2018 to December 2020 TS9. The vertical bars are the averaged $1\sigma$ retrieval errors. Right: scatterplot between 735–758 and 743–758 nm retrievals for CF < 0.2. The coefficient of determination $R^2$ and the linear regression equation are provided.

spheric absorption is also reported in Parazoo et al. (2019). Regarding potential canopy-dependent retrieval biases, we have observed that the 735–758 nm retrieval produces lower-quality fits ($\chi_r^2 > 2$) much more often than the 743–758 nm fitting window. This is especially the case for densely vege-tated areas with fully developed plants, such as the US Corn Belt in July. This effect accounts for the lower density of 735–758 nm SIF points in the Coles corn field during the summer season (first panel of Fig. 8). The reason might be in the limitation of the retrieval forward model in represent-

ing the steep reflectance spectra for high LAIs. Further tests are needed to fully characterize and solve this issue.

## 3.3 Comparison to Caltech's TROPOMI and OCO-2 products

The quality of the TROPOSIF SIF product has been investi-gated through the comparison to the Caltech TROPOMI SIF product (Köhler et al., 2018) (hereinafter, "Caltech") and SIF retrievals derived from OCO-2 (Sun et al., 2018). Using the latter products as benchmark datasets, TROPOSIF SIF data

have been evaluated in terms of absolute SIF values, systematic errors (biases), and random errors (noise).

Comparisons between the TROPOSIF (743–758 nm), Caltech TROPOMI, and OCO-2 SIF estimates were performed at the global scale for the evaluation of SIF absolute values. The instantaneous SIF retrievals were binned at 0.1° and daily resolutions, with a cloud fraction below 0.2 for those derived from TROPOMI observations. The pixel-to-pixel comparison between each pair of products shown in Fig. 9 highlights their good agreement. Over the entire year 2019, the highest agreement is found between the two TROPOMI products (TROPOSIF and Caltech), with the highest correlation (greater than 0.93) and a root mean square deviation of about $0.09 \, \mathrm{mW \, m^{-2} \, sr^{-1} \, nm^{-1}}$, while the root mean square deviation between either Caltech or TROPOSIF with OCO-2 is $0.17 \, \mathrm{mW \, m^{-2} \, sr^{-1} \, nm^{-1}}$.

The distribution of the differences between Caltech and TROPOSIF data is slightly skewed towards negative values, indicating that TROPOSIF estimates tend to be greater than Caltech SIF ones. The typical bias between TROPOSIF and Caltech is about $0.008 \, \mathrm{mW \, m^{-2} \, sr^{-1} \, nm^{-1}}$ (for the daily-corrected product). Both TROPOSIF and Caltech estimates are slightly lower than OCO-2 at 740 nm data: the typical bias between TROPOSIF and OCO-2 is about $0.029 \, \mathrm{mW \, m^{-2} \, sr^{-1} \, nm^{-1}}$, which is lower than the mean bias between Caltech and OCO-2 ($0.036 \, \mathrm{mW \, m^{-2} \, sr^{-1} \, nm^{-1}}$). The difference between the Caltech and TROPOSIF SIF estimates is due not only to the difference in the estimation algorithm, but also to the different cloud products that are used as input: concomitant cloud fraction data from TROPOMI's L2 Cloud product for TROPOSIF, when Caltech relies on cloud data derived from VIIRS (Köhler et al., 2018).

The existence of potential biases in the TROPOSIF SIF product has firstly been investigated using retrievals over large non-vegetated areas where it can be assumed that no SIF emission is present (i.e., the Sahara, Antarctica, and Greenland). The analysis has been conducted for the TROPOSIF products inferred from the two fitting windows, as well as for the Caltech SIF product and for SIF retrievals derived from OCO-2, which both act as benchmark datasets. For each SIF product, the retrievals for year 2019 were binned at 0.1° at a daily scale, screening measurements with cloud fraction above 0.05, and extracted over the regions of interest. OCO-2 SIF data were scaled at 740 nm using the approach proposed by Köhler et al. (2018). The distributions of the different SIF product values agree well over the regions considered, albeit a lesser consistency of the OCO-2 data with the TROPOSIF and Caltech estimates is observed over Greenland. It is attributed to fewer available OCO-2 observations (hence a lower decrease in the random error for each binned data compared to TROPOMI products and an increased mismatch in spatiotemporal sampling between OCO-2 and TROPOMI observations). Over the Sahara, where a higher number of measurements are available, we estimated the mean yearly bias for all products.

The mean bias for the various TROPOMI SIF estimates is slightly negative: $-0.006 \, \mathrm{mW \, m^{-2} \, sr^{-1} \, nm^{-1}}$ for the daily-corrected TROPOSIF data derived from the 735–758 nm window ($-0.017$ for the raw SIF data), $-0.028$ ($-0.080$) for the 743–758 nm TROPOSIF product, and $-0.036$ ($-0.105$) for the Caltech product. The mean bias for OCO-2 was found to be slightly positive: $0.010$ ($0.031$). All SIF products exhibit a seasonal variation of the mean bias over the different regions (note there are no OCO-2 observations over Antarctica). Over the Sahara, the magnitude of these seasonal variations is about $0.1 \, \mathrm{mW \, m^{-2} \, sr^{-1} \, nm^{-1}}$. The origin of this seasonality is still unclear, but it could partly be attributed to directional effects. We discarded that it was the result of a temporal drift by checking that the level of the SIF biases estimated for June 2018 and June 2020 was similar to the one estimated for June 2019.

We completed this analysis by characterizing the distribution of the SIF estimates from TROPOSIF and Caltech at a set of Pseudo-Invariant Calibration Sites (PICS). These sites are selected because of their radiometric temporal stability, spatial homogeneity, and high reflectance and are used for vicarious calibration of spaceborne sensors. Ten sites were considered: the six endorsed by the CEOS/WGCV/IVOS as calibration/validation test sites (Algeria3, Mauritania1, Libya4, Mauritania2, and Algeria5, which where identified by Cosnefroy et al., 1998) plus four new PICS located in Algeria, Saudi Arabia, Namibia, and Sudan (Bacour et al., 2019a). The distribution of the daily SIF values for the period 2018–2020 over these sites (data within the vicinity of 0.1°) is seen in Fig. 10 for the Caltech product and the two TROPOSIF products. The mean bias is close to $0 \, \mathrm{mW \, m^{-2} \, sr^{-1} \, nm^{-1}}$ for the three products. The smaller width of the histogram for the TROPOSIF product from the 735–758 nm window is due to the smaller single-retrieval errors. No temporal drift was detected in the SIF 2018–2020 time series for any of the sites.

Over the Sahara, we quantified the averaged value of the $1\sigma$ retrieval error (square root of the quadratic mean) for the two TROPOSIF products and the Caltech retrievals (0.1°/daily). The retrieval error for TROPOSIF is typically $0.5 \, \mathrm{mW \, m^{-2} \, sr^{-1} \, nm^{-1}}$ for the 743–758 nm fitting window estimates and $0.4 \, \mathrm{mW \, m^{-2} \, sr^{-1} \, nm^{-1}}$ for the product derived from the 735–758 nm window. These empirical values are consistent with the theoretical retrieval errors, although on the higher end. We found an typical retrieval error of $0.6 \, \mathrm{mW \, m^{-2} \, sr^{-1} \, nm^{-1}}$ for the Caltech product.

## 3.4 Evaluation of reflectance-based indices

One of the assets of the TROPOSIF product is the 665–785 nm spectral reflectance spectra attached to the SIF retrievals. Spectral reflectance data can be used to derive empirical indices and biophysical variables which are complementary to SIF, as discussed earlier in this paper.

Please note the remarks at the end of the manuscript.

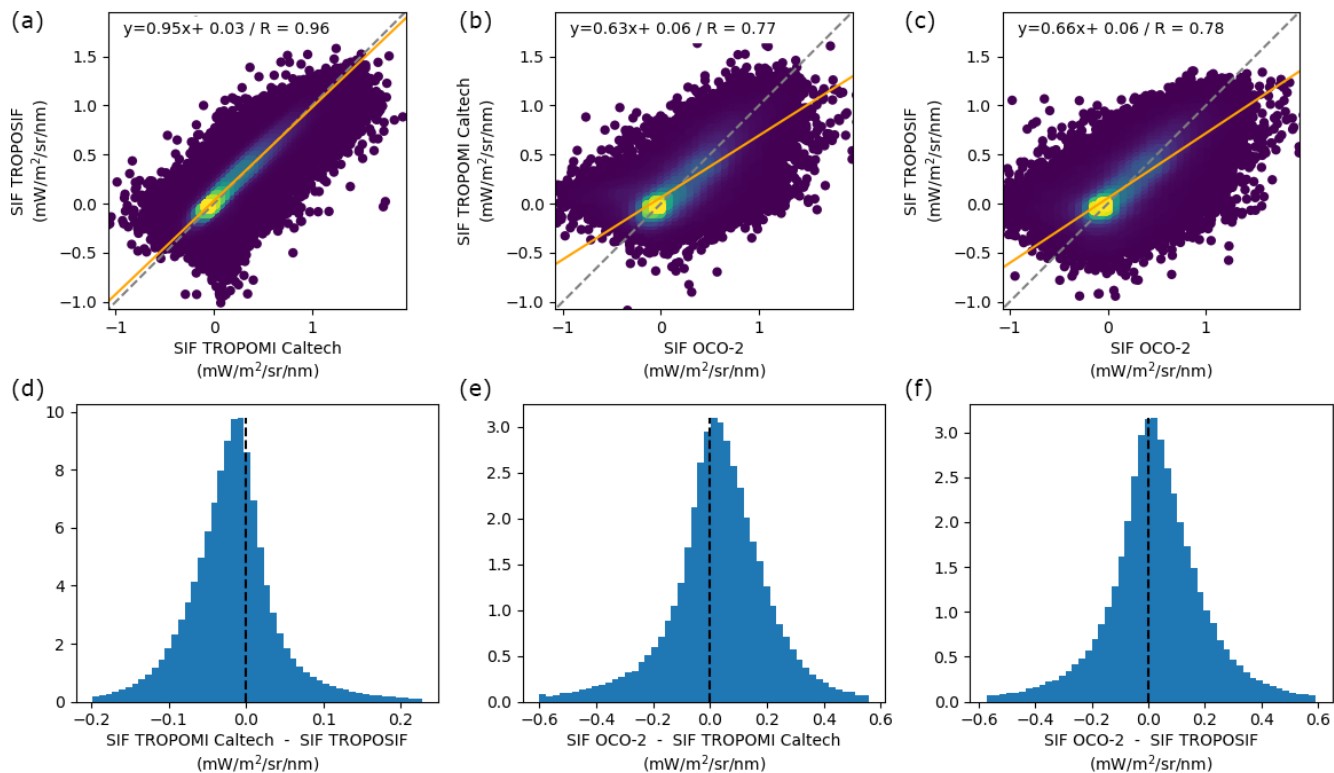

**Figure 9.** Scatterplots **(a–c)** and histograms **(d–f)** of the difference between each pair of SIF datasets, namely TROPOSIF (743–758 nm window), Caltech TROPOMI, and OCO-2 for August 2019 (daily/0.1° data, CF < 0.2). The linear regression line between two datasets is shown in orange. The linear regression equations and the correlation coefficient $R$ are provided. A day-length correction has been applied to all three datasets.

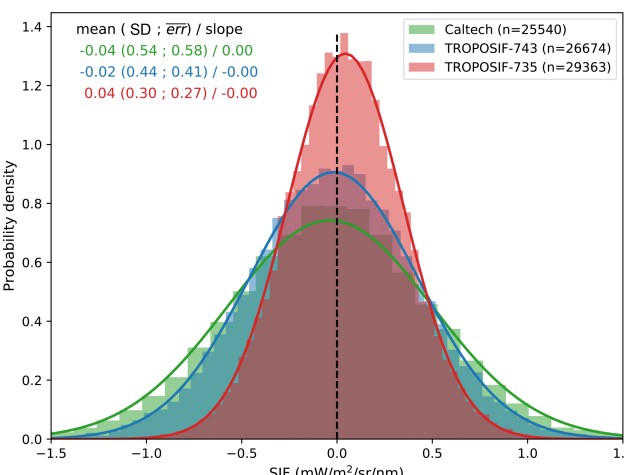

**Figure 10.** Histogram of all available TROPOMI-based SIF estimates at 10 PICS over the 2018–2020 period, for Caltech SIF and TROPOSIF SIF estimates from the 743–758 and 735–758 nm fitting windows. The distributions were fitted using a Gaussian function; the corresponding mean (bias) and standard deviation (SD TS10) are provided for each SIF product. The mean slope of the SIF temporal variations over the 10 PICS is given in brackets.

Global composites of two of such empirical indices (NDVI and NIRv) for the 8–15 July 2019 time period are shown in Fig. 11. NDVI is calculated using the 665 and 781 nm channels as red and near-infrared bands, respectively. NIRv is calculated as the product of NDVI and near-infrared reflectance (781 nm channel) (Badgley et al., 2017). The comparison of the spatial patterns in these two maps with those of SIF in Fig. 6 confirms the high correlation between SIF and NIRv that has already been reported in recent studies (Badgley et al., 2017; Zeng et al., 2019; Mengistu et al., 2021). In contrast, the dynamic range of NDVI appears to be smaller than that of NIRv and SIF, since several densely vegetated areas around the world reach the highest NDVI values.

These patterns are further illustrated in Fig. 12, which shows a comparison between SIF and vegetation indices for an ensemble of 0.5° pixels representative of several biomes. The pixel selection is described in Bacour et al. (2019b). The NIRvP is added to the analysis. NIRvP is calculated as the product of NDVI and the average TOA radiance in the SIF retrieval fitting window (also attached to the TROPOSIF product), which is a good proxy for the product of PAR and near-infrared reflectance in the original NIRvP formulation (Dechant et al., 2021). NIRvP is designed so that it mimics SIF dependencies on FAPAR, PAR, and canopy escape frac-

Please note the remarks at the end of the manuscript.

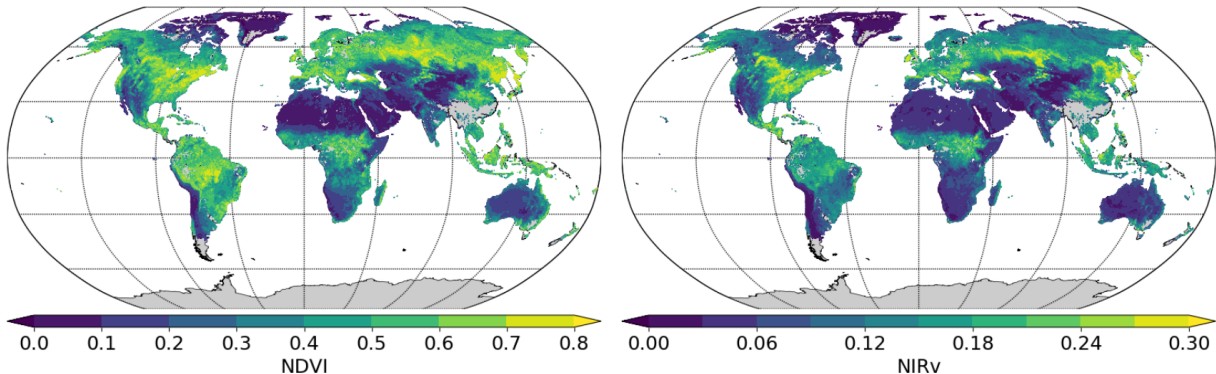

**Figure 11.** Global composites of NDVI and NIRv derived from the TROPOSIF product for the 8–15 July 2019 time period. Data are gridded in 0.2° grid boxes. A cloud fraction threshold of 0.1 is applied for cloud screening.

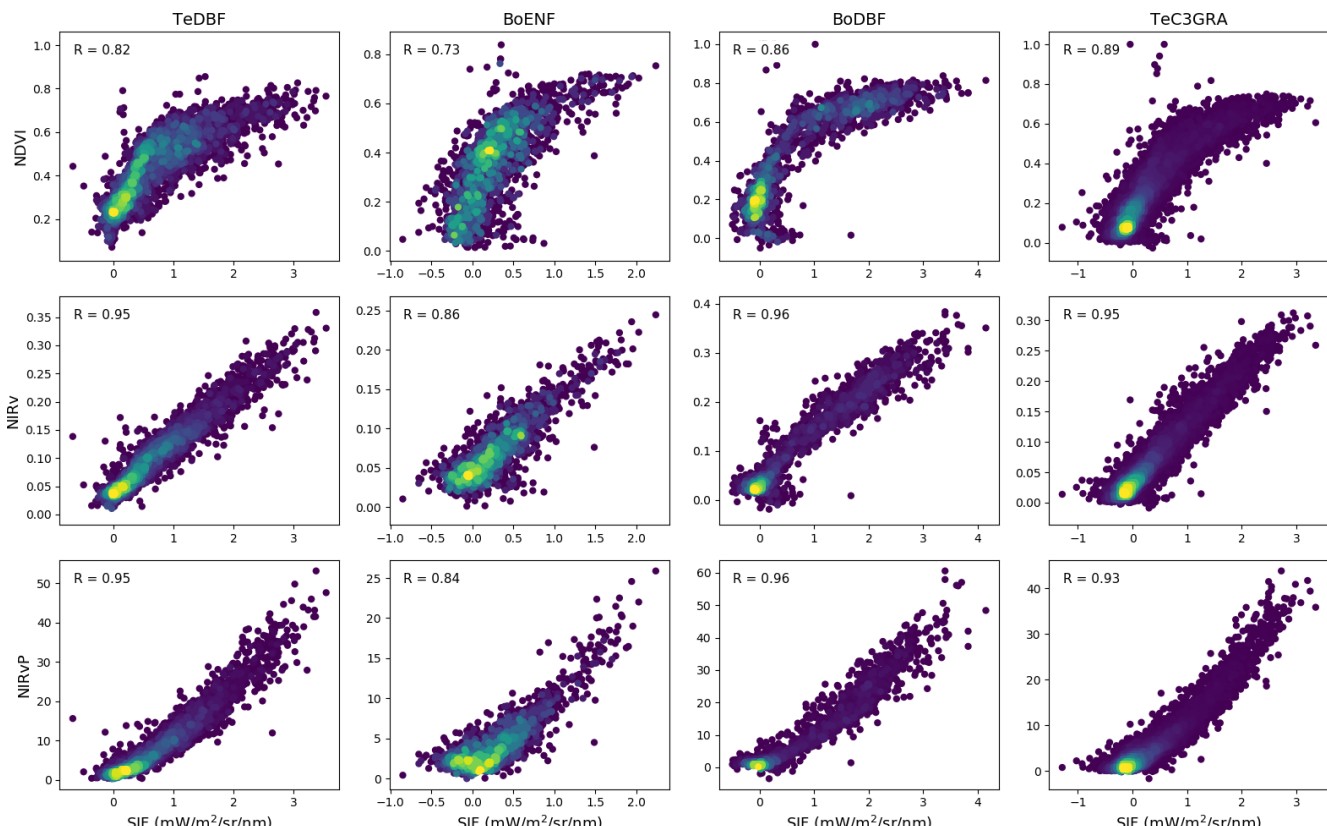

**Figure 12.** Comparison of weekly SIF data (743–758 nm, cloud fraction < 0.2) with NDVI, NIRv, and NIRvP derived from TROPOSIF spectral reflectance estimates for an ensemble of 0.5° pixels representative of temperate deciduous broadleaf forest (TeDBF), boreal evergreen needleleaf forest (BoENF), boreal deciduous broadleaf forest (BoDBF), and temperate $C_3$ grass (TeC3GRA) biomes.

tion, whereas NDVI would only indicate FAPAR, and NIRv would indicate the product between FAPAR and escape fraction CE4. The different plots in Fig. 12 indeed illustrate that SIF compares better to NIRv and NIRvP than to NDVI, with the latter saturating with respect to SIF. This effect is especially visible for the boreal deciduous broadleaf forest biome (third column of Fig. 12). On the other hand, a slightly more linear relationship is found between SIF and NIRv than be-

tween SIF and NIRvP despite the theoretically higher compatibility between SIF and NIRvP (Dechant et al., 2021). Further exploration of these relationships will be subject of future work, since the identification of practical approaches to disentangle structural and physiological contributions to the SIF signal is key to the proper use of SIF data.

## 4 Discussion

This paper has described the first version of a processing chain for the generation of far-red SIF product from the S5P-TROPOMI mission which has been developed within the framework of the ESA TROPOSIF project.

The retrieval is based on a linear forward model fitting TOA radiances in the far-red spectral region. It relies on a data-driven forward model similar to the one initially proposed by Guanter et al. (2012) for GOSAT, later adapted to GOME-2 by Joiner et al. (2013), and finally applied to simulated and real TROPOMI data by Guanter et al. (2015) and Köhler et al. (2018), respectively. In these data-driven approaches, a series of orthogonal spectral vectors derived from a so-called training set are used to model high-spectral-frequency features in the spectrum from both solar and atmospheric lines. SIF is estimated through the inversion of the linear forward model, together with the weights of the singular vectors and the coefficients of a third-order polynomial used to model low-spectral-frequency variations in the spectrum, such as those from varying surface reflectance.

Two fitting windows are selected for the retrieval, 743–758 and 735–758 nm. The first has shown to be very robust against atmospheric effects (especially cloud contamination), whereas the second gives smaller precision errors due to the greater number of spectral points. The 743–758 nm window has been selected as the baseline retrieval window, as it provides the best compromise between retrieval precision and sensitivity to clouds. This fitting window is consistent with the one used for the existing Caltech SIF product (Köhler et al., 2018), which allows a direct comparison between the two datasets. In addition, we have decided to include retrievals from the 735–758 nm fitting window as well. Retrievals from this fitting window have shown a higher sensitivity to cloud contamination but also lower precision errors because of the higher number of spectral points in the fitting window. Retrievals from the 735–758 nm fitting window can thus be advantageous for those applications using only clear-sky data.

The first assessment of the consistency of the TROPOSIF SIF data has relied on the comparison with the accurate SIF retrievals from the OCO-2 mission and with the well-established TROPOMI SIF product from Caltech. A very high similarity has been found between the two TROPOMI SIF datasets ($R$ usually greater than 0.93, 0.008 mW m$^{-2}$ sr$^{-1}$ nm$^{-1}$ bias), whereas the evaluation of precision errors over sand deserts and PICS has shown slightly smaller random errors for the TROPOSIF product. The retrieval error is typically 0.5 mW m$^{-2}$ sr$^{-1}$ nm$^{-1}$ in the 743–758 nm fitting window and 0.4 mW m$^{-2}$ sr$^{-1}$ nm$^{-1}$ in the 735–758 nm window. The mean bias for the raw SIF estimates are $-0.080$ ($-0.017$) mW m$^{-2}$ sr$^{-1}$ nm$^{-1}$ in the 743–758 nm (735–758 nm) fitting window.

In addition to SIF, spectral TOA reflectances at seven spectral points in atmospheric windows within the 665–781 nm window of TROPOMI's channels 5 and 6 are derived and included as metadata in the output files. These reflectance data can be useful to derive spectral vegetation indices and biophysical variables that can help interpret and exploit SIF data. This has been illustrated in this work by the comparison of SIF data to NDVI, NIRv, and NIRvP indices, which has shown the different information content carried by each of them.

## 5 Data availability

The dataset is available through the following digital object identifier (DOI): https://doi.org/10.5270/esa-s5p_innovation-sif-20180501_20210320-v2.1-202104 (Guanter et al., 2021). Additional informative materials are also provided on the project website at https://s5p-troposif.noveltis.fr/ TS11.

## 6 Conclusions

The TROPOSIF product presented in this document is intended to become a reference data stream for scientific work dealing with the global monitoring of vegetation, especially in those studies focused on the dynamics and productivity of terrestrial ecosystems.

Our evaluation of the TROPOSIF product in this work has shown that it has an overall similar quality to the well-established Caltech TROPOMI SIF product (Köhler et al., 2018). Two main additional features are included in TROPOSIF: a secondary SIF dataset from the 735–758 nm fitting window with an enhanced signal-to-noise ratio and spectral reflectance spectra. In the case of 735–758 nm SIF retrievals, however, the first analysis of the data has found potential difficulties of the retrieval to deal with spectrally steep radiance spectra, such as in the case of some crops. This issue will be investigated during future TROPOSIF consolidation activities. Users are recommended to use the baseline 743–758 nm SIF product for the moment. Approaches for the combination of the two SIF data streams into a single one will also be evaluated in future research.

Feedback from the user community after the first experience with the data is also expected to guide future developments. In this respect, the TROPOSIF processing chain is being implemented in the Sentinel-5P Product Algorithm Laboratory (S5P-PAL)[3], which is an ongoing project funded by the European Commission to allow fast and cost-efficient Sentinel-5P prototype product development. It is expected that the TROPOSIF product will be generated and distributed to users by S5P-PAL in an operational manner in the near future.

---

[3]https://maps.s5p-pal.com/ TS12

Please note the remarks at the end of the manuscript.

## Appendix A: L2 and L2B file description

The NetCDF4 data files containing the TROPOSIF product (either L2 or L2B) are structured as follows:

1. `METADATA/ALGORITHM_SETTINGS`, which contains information about the processor's configuration variables, such as the spectral fitting window or the thresholds used for data filtering;

2. `PRODUCT`, which contains the SIF product itself (`SIF`), the daily-average corrected SIF (`SIF_Corr`), and the estimated $1\sigma$ uncertainty (`SIF_ERROR`) for the two fitting windows;

3. `PRODUCT/SUPPORT_DATA/DETAILED_RESULTS`, which contains ancillary data fields for the exploitation or interpretation of the SIF retrievals, such as day-length correction factor (`DayLength_fac`), the reduced $\chi^2$ ($\chi_r^2$) of the fit (`redCHI2`) (in the case of L2 files), the average TOA radiance in the fitting window (`TOA_RAD`), the quality value (`QA_value`), and the spectral TOA reflectance and corresponding wavelengths at several spectral points (`TOA_RFL` and `WVL_RFL`) (in the case of L2 and L2B clear-sky products);

4. `PRODUCT/SUPPORT_DATA/GEOLOCATIONS`, which contains data fields describing the acquisition location (latitude and longitude bounds for each retrieval) (L2 files) and geometry (viewing and solar angles);

5. `PRODUCT/SUPPORT_DATA/INPUT_DATA`, which contains geospatial data used as input for the retrieval, such as the cloud fraction from TROPOMI's L2 product (`cloud_fraction_L2`) and the land cover mask used to separate land and water pixels (`LC_mask`).

**Author contributions.** Conceptualization and Methodology: LG and CB Formal analysis: LG, CB, AS, TAvK, FM, PK Investigation: All authors. Resources: LG, CB, AS, FM, Supervision: LG, CB, IA, CR, Writing-original draft: LG and CB, Writing-review and editing: All authors. TS13

**Competing interests.** The contact author has declared that neither they nor their co-authors have any competing interests.

**Disclaimer.** Publisher's note: Copernicus Publications remains neutral with regard to jurisdictional claims in published maps and institutional affiliations.

**Acknowledgements.** The TROPOSIF project is funded by ESA's Sentinel-5p+ Innovation activity (ESA contract no. 4000127461/19/I-NS). Christian Frankenberg and Philipp Köhler acknowledge funding through NASA's Earth Science U.S. Participating Investigator grant NNX15AH95G. The authors thank Léo Grignon (NOVELTIS) for the development of the TROPOSIF website (https://s5p-troposif.noveltis.fr/ TS14), Alexandru Dandocsi from ESA for his support to obtain a DOI for the dataset, and Elena Sánchez-García (UPV) for her earlier data quality assessment.

**Financial support.** This research has been supported by the European Space Agency (grant no. 4000127461/19/I-NS) and the National Aeronautics and Space Administration (grant no. NNX15AH95G). TS15

**Review statement.** This paper was edited by David Carlson and reviewed by two anonymous referees.

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

## Remarks from the language copy-editor

CE1 Please note the slight edits to affiliations 4 and 7.

CE2 Please note that in keeping with the dominant style of this manuscript, it has undergone copy-editing according to the standards of American English.

CE3 In keeping with the dominant style of the manuscript, no serial (Oxford) commas were removed.

CE4 The initial sentence had to be rephrased; please check and confirm or offer an alternative.

## Remarks from the typesetter

TS1 Please provide department.

TS2 Please provide last access date.

TS3 Please check throughout the text that all vectors are denoted by bold italics and matrices by bold roman.

TS4 Please confirm percent signs.

TS5 Is this a citation? If so, please provide reference list entry.

TS6 Please provide last access date.

TS7 The composition of Figs. 4, 5, 7, 8, 9 and 10 has been adjusted to our standards.

TS8 Figs. 5, 7 and 9 are labeled now. Please also check the caption.

TS9 Please confirm date.

TS10 Please note that "std" has been changed to "SD" according to our standards.

TS11 Please provide a direct link to the data set and, if possible, a DOI instead of a URL. In any case, please provide a reference list entry including creators, title, and date of last access.

TS12 Please provide last access date.

TS13 It should be clear who contributed to which part of the manuscript. For example, there are guidelines (see https://publications.copernicus.org/for_authors/obligations_for_authors.html) for who may be listed as a co-author. Please provide us with a more specific text (in complete sentences) for the Author contributions section.

TS14 Please provide last access date.

TS15 Please note that the funding information has been added to this paper. Please check if it is correct. Please also double-check your acknowledgements to see whether repeated information can be removed or changed accordingly. Thanks.

TS16 Please ensure that any data sets and software codes used in this work are properly cited in the text and included in this reference list. Thereby, please keep our reference style in mind, including creators, titles, publisher/repository, persistent identifier, and publication year. Regarding the publisher/repository, please add "[data set]" or "[code]" to the entry (e.g. Zenodo [code]).

TS17 Please provide page range or article number.

TS18 Please provide page range or article number.

TS19 Please provide publisher or journal information.

TS20 Please note that the information "[data set]" has to appear next to the repository. Please confirm.

TS21 Please provide all author names.

TS22 Please provide page range or DOI.

TS23 Please provide page range or article number.

TS24 Please note: Article and year was updated.

TS25 Please note: Article DOI and year was updated.

TS26 Please provide page range or article number.

TS27 Please provide page range or article number.

TS28 Please provide page range or article number.

TS29 Please provide all author names.

TS30 Please provide page range or DOI.