# Peer review of "The TROPOSIF global sun-induced fluorescence dataset from the Sentinel-5P TROPOMI mission"

_Earth System Science Data, 2021_

## Author Comment (AC1)

**Reviewer 1**

**General comments:**

The researchers accomplished SIF retrieval from the Sentinel-5P TROPOMI mission and provided data with high quality, which expands the application of TROPOMI data in vegetation monitoring. The methods and materials used in the manuscript are reasonable and described in detail, which can support the publication of the dataset. The dataset is accessible and complete, the quality value and retrieval error of the data were fully evaluated. The TROPOSIF product has a high consistency compared with previous SIF products, and the results are reliable. There are only a few questions to be discussed.

Thank you for the positive comments.

**Specific comments:**

1. Line 37 to 42: Since Caltech's TROPOMI SIF product has been proven to be effective, the gaps in previous research and the purpose of this research should be more clearly stated.
   The following text has been added (**L63-66**):
   "Our work is aimed at developing a TROPOMI-based SIF processor which can be implemented at ESA's data processing facilities for the operational generation and distribution of the data product to users. In addition to SIF, reflectance spectra from each input radiance spectrum are also included in the product for combination with the SIF retrievals."

2. Line 88 to 90: To what extent can the influence of the atmosphere be considered negligible? Can you add a comparison to compare the retrievals using calculated effective atmospheric transmittance and transmittance set as 1?
   The following text has been added (**L96-99**):
   The effect of atmospheric absorption on SIF retrievals at far-red wavelengths had been previously evaluated by means of simulation in \citet{Guanter_SVD_2012, Frankenberg_2012}. The effect should be in the range $\sim$3--6\% for a typical aerosol optical thickness of 0.2 and observation angles between 0$^\circ$ and 45$^\circ$.

3. Figure 8: Previous SIF products have shown a tendency that SIF magnitudes decrease with narrower fitting windows toward longer wavelengths near the far-red fluorescence peak and in fitting windows with less water vapor absorption (Parazoo, 2019), which is inconsistent with the results shown in Figure 8, how do you account for this?

   In our case, retrievals in both fitting windows are normalized to 740 nm. For that, a fixed TOC SIF spectrum is used. We find variations in the slope of SIF(743-758) = f(SIF(735-758)) between 1 and 1.5. We argued in the text that "Further research is needed to understand whether the variations in the slopes are due to retrieval biases over some vegetation types or to leaf/canopy radiative transfer effects making the shape of the SIF emission to depend on the leaf and canopy type."

   This text referring to previous results by Parazoo et al. has been added "A dependence of SIF retrievals on the fitting window and atmospheric absorption is also reported in \citet{parazoo_2019}" (**L283-285**)

Parazoo, N. C., Frankenberg, C., Köhler, P., Joiner, J., Yoshida, Y., Magney, T., Sun, Y., and Yadav, V.: Towards a Harmonized LongTerm Spaceborne Record of Far-Red Solar-Induced Fluorescence, Journal of Geophysical Research: Biogeosciences, 124, 2518–2539.

1. Line 393 to 398: The use of the 735-758nm fitting window is a feature of this research, but the limitations of this window were also stated. Is it possible to select one of the retrievals from the two fitting windows for each observation according to several indicators (e.g. cloud fraction threshold) to merge the retrievals from the two fitting windows and maintain the advantages of both, rather than providing two separate datasets?

   Thanks for the suggestion, this is a very interesting idea. The main limitation to combine the two SIF retrievals in the way that you describe might be in the fact that they might be differently affected by retrieval biases and/or radiative transfer effects, so that the combination of the two data sets might introduce noise in the time series with respect to that consisting in one single data set.
   We have added the following text to the manuscript "Approaches for the combination of the two SIF data streams into a single one will be evaluated in future research" (**L411**)

**Technical corrections:**

1. Figure 1: The "FT" in the figure note is inconsistent with the abbreviation "FW" in the figure.
   Caption corrected.
2. Figure 2: Only the weights of the first 8 singular vectors are shown in the figure, which is inconsistent with the figure note.

   Caption corrected.

---

## Author Comment (AC2)

**Reviewer 2**

The paper describes a global dataset of Sun-Induced chlorophyll Fluorescence (SIF) obtained by the Sentinel 5P-TROPOMI mission. I believe this dataset is of great value to the community, and I want to express my gratitude to the authors for their diligent work.

Thanks to you for this positive feedback.

L28: In addition to the GPP related studies, I suggest mentioning also the value of SIF for hydrological studies, e.g., on ecosystem transpiration and water limitation (see, e.g., Maes et al. Remote Sensing of Env. 2020; the review of Jonard et al. Agric. Forest Meteo. 2020; Gonsamo et al., Remote Sensing of Env. 2019).

Done. We have added references as "and also ecosystem transpiration and water limitation (Pagán et al., 2019; Maes et al., 2020; Shan et al., 2021)." (**L29**)

L50-55: It might be interesting to mention the Fluorescence Correction Vegetation Index by Yang et al. Remote Sensing of Env. 2020 here. This is, similar to NIRv, an index to assess the effect of canopy structure. I'm not entirely sure whether their framework is applicable to TROPOMI data, but it's certainly worth mentioning.

Thanks, a reference to Yang et al. (2020) has been added. (**L51**)

L54-56: I have a few annotations to make on the sentence about SIF yield, used for monitoring vegetation stress. First, it is important to highlight the difference between SIF yield and fluorescence yield. SIF yield is a canopy-scale variable, affected by both canopy structural and leaf biochemical aspects. SIF yield is relatively easy to retrieve from satellites. The variable that is however directly impacted by a plant's stress status is the photosystem scale fluorescence quantum yield. This variable is fundamentally different from SIF yield as it does not depend on variables such as chlorophyll content, leaf area index, or leaf orientation. Measuring the latter variable from satellite imagery is not self-evident. Celesti et al. Remote Sensing of Env. 2018 provides a framework for retrieving the fluorescence quantum yield at the canopy scale. Please clarify how you define the SIF yield and how it can be compared to photosynthetic efficiency?

L56: I understand the point that you want to make here, but I suggest being more careful in the wording. While Dechant et al. indeed showed that NIRv is a better predictor for GPP compared to SIF, I am not sure to which extent this idea holds in case of a stress situation. SIF can be decomposed into the following factors:  $SIF = PAR \cdot fPAR \cdot Phi\_f \cdot f\_esc$

NIRvP only takes into account $PAR \cdot fPAR \cdot f\_esc$. This does not consider variations in Phi_f despite the latter's sensitivity to short-term stresses. This is why Dechant et al. raises the point that NIRvP might show significant disagreements with SIF in case of short-term stresses, such as droughts or heatwaves. We believe that NIRvP bears the potential to serve as a 'potential SIF emission' or 'reference SIF emission' (similar to reference evapotranspiration using the Penman-Monteith method). The ratio between NIRvP and SIF could then serve as a stress factor, as it more or less isolates Phi_f. It is however worth noting that there is not yet any experimental evidence supporting this claim, despite the seemingly simple logic behind it.

Thanks for these two clarifications related to the interpretation of NIRv/NIRvP and their potential for the calculation of SIF yield (L54-56). We agree with the distinction between SIF and fluorescence yield. This part has been rewritten as "NIRv and NIRvP could then provide useful information for the calculation of the SIF yield, which is a top-of-canopy variable providing information on both fluorescence yield (leaf-level variable) and any remaining canopy structure effects not accounted for by NIRv and NIRvP." (**L56-58**)

L80: It would be good if the authors could provide a brief explanation of the meaning of a and α.

Information added.

L279: Section 3.3 provides an interesting comparison between SIF products. It describes a difference in the SIF signal depending on which bands have been used for the retrieval. It describes that OCO-2 and Sentinel 5P-TROPOMI data are comparable. I wonder to which extent the SIF data from the dataset here will be comparable to FLEX-derived SIF data, as it will not be based on the exact same bands. Do you have an idea on that?

Thanks for the thoughts on the upcoming FLEX mission. As far as we know, the FLEX retrieval will rely on oxygen absorption bands centered at 687 and 760 nm. The second one is relatively close to the far-red window used in our retrieval, with a 740 nm reference wavelength, so the comparison should be feasible.

However, we do not consider this discussion to be central to our manuscript and have opted for not performing any action on the manuscript in response to this comment.

E.g. Figure 9 and text below: Information on the interpretation of the negative SIF values is missing. Could you please explain the reason why there are negative SIF values and what their physical meaning is? It would be good to know for future users whether these negative SIF data should be considered or not?

This clarification has been added: "On the other hand, negative SIF values are found for the two fitting windows. These are caused by noise in the data being propagated to random errors in the retrieved SIF. Removing negative SIF retrievals from the processing should be avoided, as it would lead in positively-biased averages." (**L278-280**)